# A vehicle license plate data access model based on the jump hash consistency algorithm

**Wei Wang**[1], **Wenfang Cheng**[2]*, **Jing Chen**[1], **Zhen Wang**[3], **Yuran Zhang**[3], **Yingfang Yu**[4]

**1** College of Information Technology Engineering, Tianjin University of Technology and Education, Tianjin, 300391, China, **2** Polar Research Institute of China, Ministry of Natural Resources of the People's Republic of China, Shanghai, 200136, China, **3** TianJin Survey Design Institute Group Co., Ltd, Tianjin, 300191, China, **4** China-Singapore Tianjin Eco-city Technology Innovation Bureau, Tianjin, 300480, China

* chengwenfang@pric.org.cn

**Data Availability Statement:** All relevant data are within the paper and its Supporting Information files.

**Funding:** W W, C J. 20220105 Tianjin Jinnan District Bureau of Science and Technology http://

## Abstract

The massive amount of vehicle plate data generated by intelligent transportation systems is widely used in the field of urban transportation information system construction and has a high scientific research and application value. The adoption of big data platforms to properly preserve, process, and exploit these valuable data resources has become a hot research area in recent years. To address the problems of implementing complex multi-conditional comprehensive query functions and flexible data applications in the key–value database storage environment of a big data platform, this paper proposes a data access model based on the jump hash consistency algorithm. Algorithms such as data slice storage and multi-threaded sliding window parallel reading are used to realize evenly distributed storage and fast reading of massive time-series data on clustered data nodes. A comparative analysis of data distribution uniformity and retrieval efficiency shows that the model can effectively avoid generating the cluster hotspot problem, support comprehensive analysis queries with various complex conditions, and maintain high query efficiency by precisely positioning the data storage range and utilizing parallel scan reading.

## 1. Introduction

With the rapid development of China's urbanization process and the continuous improvement of the population's living standards, the number of car owners in cities across the country is increasing yearly, with huge pressure to support urban road traffic amid traffic congestion that is very common in various cities [1–3]. In order to reasonably and fully utilize existing road resources, effectively improve road capacity, and significantly reduce urban traffic congestion index, the development, deployment, and application of intelligent transportation systems (ITS) have been widely carried out throughout China [4,5]. ITS is the information technology, computer technology, data communication technology, sensor technology, electronic control technology, artificial intelligence and other advanced science and technology effectively used in transportation, service control and vehicle manufacturing, strengthen the connection

www.tjjn.gov.cn/zwgk/zcwj/qjwj/qkjj100/ The funders had no role in study design, data collection and analysis, decision to publish, or preparation of the manuscript.

**Competing interests:** The authors have declared that no competing interests exist.

between vehicles, roads, users, so as to form a comprehensive transportation system. ITS plays a very important role in improving traffic management efficiency, relieving traffic congestion, reducing environmental pollution, and ensuring traffic safety. With the rapid development of big data technology, artificial intelligence, machine learning, AI and other cutting-edge technologies, ITS is developing toward vehicle-road network collaboration, accurate real-time information acquisition, and artificial intelligence-aided decision-making. Data collection, communication and management are the core technical issues of current and future ITS development and construction [6,7]. While, the traffic chokepoints spread in the traffic network are the core data collection facilities in ITS. Automated steps, such as capturing images, recognition of license plates, recording speeds, and packaging and transmitting communication modules using core HD camera equipment at traffic checkpoints, generate a huge unstructured data set (vehicle license plate data, henceforth referred to as VLPD) containing vehicle driving status and image information on a terminal server [8–10]. Since the VLPD includes rich information such as license plate numbers, the time, license plate colors, vehicle types, lanes, vehicle speeds, driving directions, and captured images, it is widely used in many applications such as traffic flow analysis, road planning, vehicle tracking, and security management [11–14]. The volume and availability of VLPD in ITS leads to the need for data-driven approaches. Big data algorithms are being applied to further enhance the intelligence of transport applications. The application of big data algorithms has received increasing attention in both the academic and industrial fields of ITS. Big data algorithms in ITS have a wide range of applications including but not limited to signal recognition, object detection, traffic flow prediction, travel time planning, travel route planning and safety [15]. Knowing how to properly store, efficiently process, and exploit these valuable VLPD resources is necessary to ensure the successful solution of urban traffic problems.

Due to the rapid growth of the VLPD volume, it is difficult for a traditional relational database to support the needs of fast data access and real-time analysis applications. Given the characteristics of big data storage platforms in terms of the horizontal scalability of the data storage capacity and handling high-throughput write operations and fast data acquisition based on primary keys, there is an industry-wide consensus that big data storage platforms should be adopted for storing, processing, and applying VLPD in order to solve the storage and application problems [16]. However, since the key–value databases hosted on big data platforms generally provide only simple data retrieval functions based on row keys, they cannot cope with complex, multi-conditional queries, which poses a huge challenge for VLPD analysis applications that are based on big data platforms [17,18]. How to solve the key technical problem of comprehensive, multi-conditional queries has become a hot research topic for addressing the problem of effective storage and application of VLPD on a big data platform.

In this paper, based on a comprehensive summary of the existing research progress and the problems arising from characteristics of VLPD such as its time-series nature and uneven distribution of data, we design a VLPD access model that ensures that data is evenly distributed on cluster nodes and supports parallel multi-threaded requests, thus providing fast access to VLPD on a big data platform and successfully addressing the problem of multi-conditional integrated traffic data queries in a key–value big data environment.

## 2. Related work

The existing research on the problem of improving the efficiency of multi-conditional integrated queries in the key–value database environment of big data platforms can be divided into several categories, which concern auxiliary indexing technology, MapReduce, and Spark.

The first area of research explores auxiliary indexing techniques to achieve multi-conditional querying of VLPD. In [19], a data storage architecture using three data tables is proposed to separately meet the three types of traffic analysis requirements. In the three tables, the big table row keys are respectively designed in the order T-C-P, C-T-P, and P-T-C to realize the functions of time-series-based querying, crossing-based querying, and license-plate-based querying. In [20], based on a detailed analysis of the underlying indexing technology of the LSM(log-structured merge)-based key–value database, a diff-index algorithm model for distributed LSM databases and support for multi-conditional queries is proposed. The diff-index is based on LSM's global design to achieve a balance between read and write performance and data consistency. In [21], based on a thorough analysis of the working principles of five representative secondary indexing methods (Eager, Lazy, Composite, Zone maps, and Bloom filters) for LSM-based key–value stores, the LevelDB++ system is constructed to implement these indexes, and guidelines are proposed for different workloads and for choosing the correct indexing techniques for different workloads and applications. In general, when auxiliary indexing techniques are used to implement multi-conditional complex query application problems in key–value databases, a combination of data tables and index tables is used for data retrieval for different applications. By prioritizing the data row keys based on the query elements, the query scanning data range is reduced and the query efficiency is improved. However, the efficiency of multi-conditional complex queries that are implemented using auxiliary indexing techniques comes at the expense of requiring large amounts of storage space.

The second area of research explores data processing technology for big data analytics applications using Apache Hadoop MapReduce, which is a programming model and algorithm for large-scale data set processing in a distributed environment. In [22], a MapReduce parallel processing framework is used to process large volumes of taxicab GPS tracking data on a Hadoop distributed computing platform for data extraction, data statistics, and data integration, which in turn enables traffic analysis functions for traffic on big data platforms. The research in [23,24] combines MapReduce techniques and deep learning to analyze big data for traffic on a Hadoop big data platform to detect driver violations. In [25,26], MapReduce is used to implement a function that can calculate the shortest path in the road network more quickly and thus achieve faster and more accurate traffic route planning. In general, MapReduce techniques are mainly used to process large-scale datasets for offline analysis. The implementations construct query conditions as the starting and ending positions of row keys and scan the range in parallel. If the range of row keys lies in multiple region servers, the MapReduce framework automatically uses the boundaries of the regions to divide the tasks and automatically execute the queries in parallel. However, the query range can only be defined once for each MapReduce task, which makes a full scan of all regions within the row key range inevitable, so that MapReduce is generally less efficient and therefore not suitable for handling business needs with high real-time requirements.

The third area of research centers on Apache Spark, a memory-based distributed computing engine. By caching a large amount of data in memory, the query speed is improved. Spark adopts the RDD (Resilient Distributed Data) concept, which can support data slicing and parallel computing. The research in [27,28] examines data foundation frameworks and data models based on Spark technology for business needs in smart city construction and urban management, introducing intelligence and refinement in urban management to realize real-time detection, analysis, and data processing in various areas of a city. In [29–31], Spark technology is used to address the traffic flow prediction problem in order to optimize traffic flow and improve road utilization and traffic efficiency. By monitoring and analyzing traffic data in real time, traffic congestion and anomalies are identified, and future traffic flows are predicted and planned. Compared to the MapReduce framework, Spark has the advantages of faster disk

IO computation, richer API programming language support, and more comprehensive application scenarios. However, Spark also has obvious shortcomings, including the need for larger memory, high network bandwidth requirements, a lack of stability, and high complexity requirements.

In summary, existing studies have explored improving the performance of the multi-conditional integrated query capabilities of key–value databases for big data platforms from various perspectives, such as the storage structure, server-side parallel processing, and in-memory stream computing. Although various research solutions have made progress on specific problems, there are certain limitations in terms of technical complexity, implementation costs, and application extensions. Given VLPD's characteristics of being massive, its organization as a time series, and its uneven distribution of data along the dimension of time, the cluster parallel computing advantage of big data platforms has not been fully exploited due to the lack of a good solution to the cluster node load balancing problem, and there is still much room for improving their performance.

## 3. VLPD storage model

In the following, we study the implementation of a data storage model with absolute load balancing characteristics, starting with the characteristics of traffic vehicle plate data.

### 3.1 Data characterization of VLPD

VLPD contains information such as license plate numbers, the time of capturing images, and the location, vehicle type, and recorded image, which are generated by the HD equipment deployed at traffic checkpoints. Traffic checkpoint data has the following important characteristics: 1) It is massive: As each vehicle passing through a traffic checkpoint generates a data record, given the thousands of checkpoints with recording equipment throughout a city, a large amount of passing traffic data is generated every moment. With the passage of time, the accumulated data volume becomes larger and larger. 2) The data forms a time-series: Traffic checkpoint data are continuously arranged by time, where the moment of the vehicle's appearance and the vehicle's unique identification information (license plate number) can be used to uniquely define information about a passing vehicle. 3) It is unevenly distributed: Influenced by various factors such as morning and evening travel peaks, holidays, and weather, the distribution of data along the time axis has an obvious unevenness.

In order to ensure fast access to large-scale license plate data on a big data platform, we first need to solve the problem of access hotspots that are caused by the temporal character and unevenness of the temporal distribution of the data.

### 3.2 Jump hash consistency algorithm

Based on the preceding analysis of the data characteristics of VLPD, the combination of the moment a vehicle passes through a certain checkpoint and the vehicle's license plate number (moment + license plate number) can be used to uniquely determine any piece of VLPD. Due to many factors such as weekday travel mornings and evening peaks, holidays, and so forth, the data is unevenly distributed in time. Based directly on the moment + license plate number combinations, it is easy to trigger data access hotspots, which result in load imbalances and seriously affect the performance of data reads and writes.

In the following, we assume that $N$ slots are preset in each cluster of a big data platform to store the VLPD and that the slots are evenly grouped in order, corresponding to the physical servers (see Fig 1).

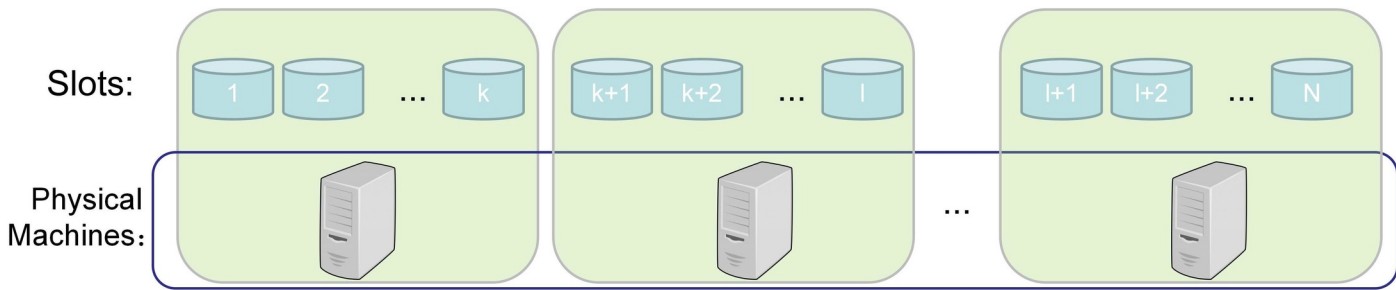

**Fig 1. Relationship of logical slots and physical machines.**

Achieving a uniform distribution of data across the physical clusters and ensuring load balancing during writes and queries is equivalent to the equal probability that each piece of VLPD is located in any one slot.

Letting **key** uniquely define a row of the VLPD and letting **n** denote the slot number where a piece of data is located, any target slot number **n** must satisfy the following equation:

$$P_n(key) = \frac{1}{N}. \tag{1}$$

That is, for any **key**, the probability that **n** is distributed in the target slot interval [0,N-1] is 1/N.

To ensure that Eq (1) holds, we require a function **f** such that:

$$n = f(key, N). \tag{2}$$

That is, for any **key**, there is a unique slot numbered **n** among the total slots **N** such that the probability of **key** being located in slot **n** is **1/N**.

Our implementation of the function **f** uses the *jump hash consistency algorithm* (JHCA) [32,33]. JHCA is a hashing algorithm for data distribution that maps keys into a defined range of integer values using a hash function, and we map the integer value to a predefined set of servers, thus achieving load balancing and efficient of data access.

Assuming that the final hash function that satisfies balancing and consistency is **ch(key, buckets)**, the core idea of the hash consistency algorithm can be described by the following recursive relation:

- When **n** = 1, all keys are to be mapped into the same bucket, i.e. **ch(key, 1) = 0**.

- when **n** = 2, **K** / 2 keys need to be mapped into each of the two buckets for uniformity (where **K** is the total number of keys), so **K** / 2 keys need to be remapped.

- . . .. . .

- When the number of buckets increases from **n** to **n** + 1, there are **K** / (**n** + 1**)** keys that need to be remapped.

Based on this algorithmic idea, the pseudo-random numbers generated by the linear congruence generator (LCG) can be used to determine which keys are remapped into new buckets [34]. As the number of buckets **n** changes from 1 to the maximum value, fewer and fewer key redistributions need to occur. That is, there is no need to determine whether to migrate the key for each value of **n**, but rather a need to jump forward, and this is done by the JCHA. This algorithm can be described using the Java language as follows.

```
Algorithm 1;: Jump Hash Consistency Algorithm
1 private static int ch(long key, int buckets) {
2     long b = -1;
3     long j = 0;
4     while (j < buckets) {
5       b = j;
6       key = key * 2862933555777941757L + 1;
7       j = (long) ((b + 1) * (double) (1L << 31) / ((double) ((key >>>
33) + 1)));
8     }
9     return (int)b;
10 }
```

In the code, the ***ch*** method receives a long integer ***key*** and an integer ***buckets***, and it returns the mapped value of the ***key*** in the ***buckets***. If the ***buckets*** are defined as the preset slots in the big data platform that are used to store the VLPD, and the ***key*** is the unique identifier of any piece of the VLPD, then the ***ch*** function's return value is the slot number where the piece of data is located.

The JHCA has the advantage of being efficient and scalable. For a given ***key***, it requires only one hash calculation to determine the mapping server for that ***key***, rather than multiple calculations like the traditional hashing algorithms. It has been theoretically and empirically shown that the time complexity of the JHCA for computing slots is ***O(ln(n))***. It also has high scalability with relatively little data movement when adding or removing servers.

### 3.3 Storage structure design

HBase is a Hadoop-based columnar storage system designed to handle large-scale unstructured and semi-structured datasets. HBase manages huge datasets as key–value tuples and uses key–value mapping to achieve highly consistent, elastic, real-time read and write access to big data. Because of its open source, mature development, and wide application in various industries, HBase has been widely used in the field of big data platforms.

The HBase database is an open-source implementation of Google's paper "Bigtable: A Distributed Storage System for Structured Data" [35]. The row key of an HBase table is the focus of the design, which determines how the data is stored and accessed. Row keys should be unique, stable, sortable, and divisible. It is also necessary to avoid row keys that are too long, too complex, or contain sensitive information. Considering many factors such as traffic car data characteristics and application requirements, the designed HBase table structure is shown in Fig 2.

**Row keys**: The row key design is the core of the HBase table structure design. The row key for storing VLPD consists of three parts: a slot number, the vehicle's time of passing the checkpoint, and the license plate number. The slot number is used as the header of the row key to

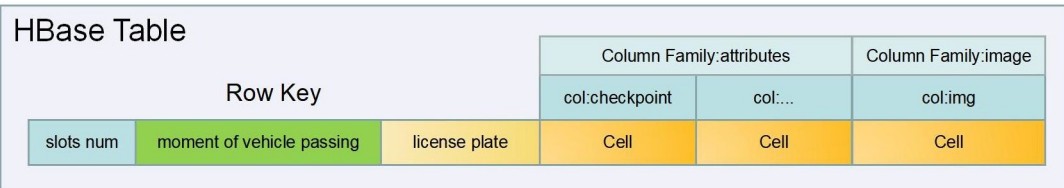

**Fig 2. Storage structure for VLPD on HBase.**

indicate where this data is located, and the combination of the time the vehicle passed through the checkpoint and the license plate number is used to uniquely identify a passing vehicle's information.

**Column families**: Each row key corresponds to two column families: One column family holds the attribute information, and the other holds the image that is captured when the vehicle passes by.

### 3.4 Row key generation algorithm

Considering the time-series nature of the VLPD and the uneven distribution of data in the time series, the VLPD as arranged in the time series is processed in minute slices. That is, a vehicle's time of passing through the traffic checkpoint expressed as a minute is converted into a timestamp that is used as the first parameter *key* for JCHA, and the number of preset slots in the big data platform is used as the algorithm's second parameter *buckets* to calculate the slot number corresponding to the vehicles passing during the minute slice. The result of the calculation is used as the row key header for all of the data in the time period, and the combination of the time of the vehicle passing through the checkpoint and the captured license plate number forms the unique row key for the data saved in the database.

The slot calculation algorithm is described by the following Java code.

```
Algorithm 2;: Row Key Generation Algorithm
1 private byte[] generateRowkey(JSONObject row) {
2     //Accurate time to the minute of the vehicle passing through the
checkpoint
3     String slotString = row.getString("JGSK").substring(0, 16) +
":00";
4     //Convert time to a long timestamp
5     long timeStamp = getTimeStamp(slotString);
6     int slotIndex = ch(timeStamp, buckets);
7     String rowKey = String.valueOf(slotIndex+10000).substring(1) +
"|"
8         + row.getString("passTime") + "|" + row.getString("plate");
9     return rowKey.getBytes(StandardCharsets.UTF_8);
10 }
```

In the code, row is the VLPD data in JSON object format, and buckets is the number of slots preset in the big data platform to save the data.

### 3.5 VLPD storage on HBase

HBase uses its core component, the region servers, to manage data access. Each region server manages multiple regions, each of which represents a partition of an HBase table. The region server manages regions according to the range of the row keys in an HBase table, and when the table data grows, HBase splits the region into two or more sub-regions in accordance with certain rules to achieve load balancing. The relationship between rows, regions, and region servers can be expressed in dictionary order by the HBase region server architecture, as shown in Fig 3.

Since there are 1440 minutes in a day, we can set the number of slots where the traffic cars' plate data is stored to the constant 1440. The slot identifiers located at the head of the row key vary continuously from 0000, 0001,. . . to 1439. The traffic car data sliced by the minute of a day will be randomly distributed in these slots with equal probability.

In HBase, pre-partitions (regions) on the server are generated using the following script:

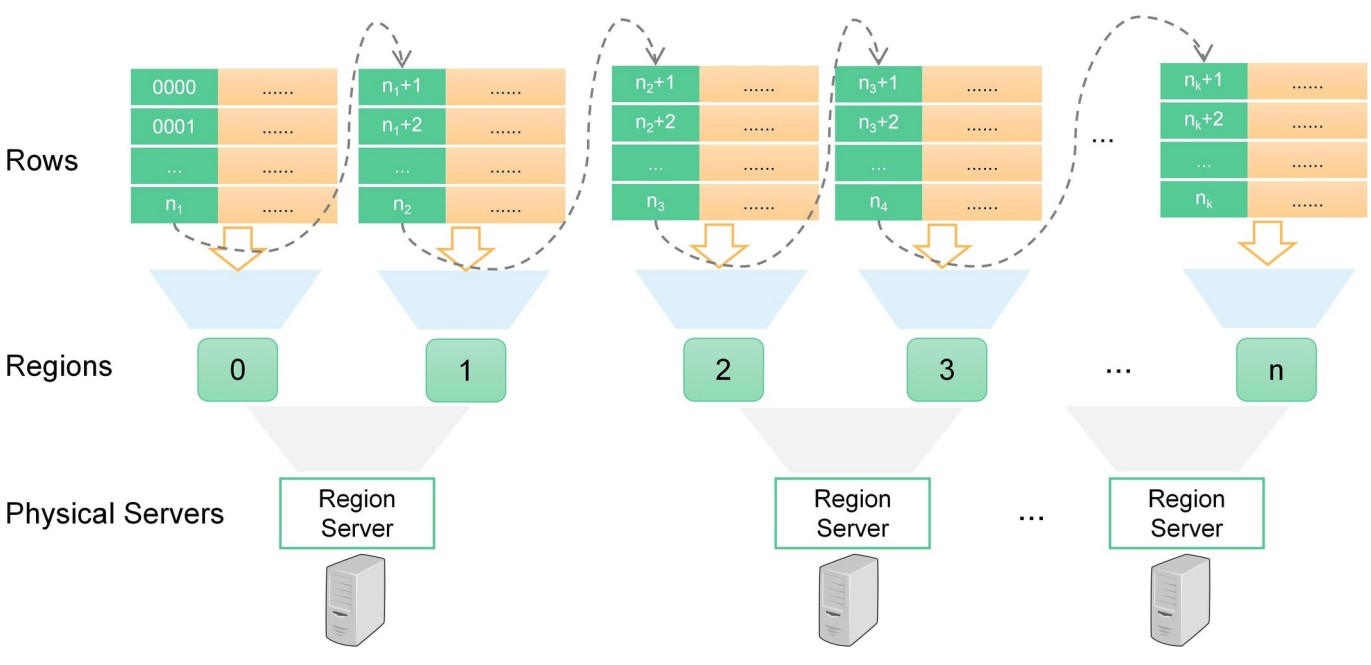

**Fig 3. HBase region server's architecture.**

```
$>create 'Traffic, {NAME=>A', NAME=>'I', VERSIONS=>1, SPLITS=>
['0060','0120','0180',...,'1380'].
```

Executing this script in the HBase shell creates a large table named "Traffic" in the database. The large table has two column families (A and I), which are used to store the attribute information and the captured images, respectively, for the passing cars. At the same time, the table is pre-partitioned to create 24 regions, and HBase will save the data to the corresponding regions based on the slot numbers located at the head of the row keys.

## 4. VLPD retrieval model

Based on the storage model, the VLPD retrieval model should solve the problem of improving the efficiency of data multi-conditional comprehensive queries. Since the time factor is a necessary condition for all VLPD applications, the query function for all comprehensive conditions within a certain time range can be realized by a data scanning request to the server initiated by the client. The data storage structure based on JHCA guarantees that the VLPD slice is uniformly and randomly distributed in the preset slots. The improvement in the data query efficiency can be considered in terms of two aspects, namely, the precise positioning of the data distribution range and the full play being given to the cluster's parallel request processing capability.

### 4.1 Data distribution range location

VLPD applications can be summarized by two categories: statistical analyses of vehicle traffic and vehicle tracking. For the statistical analysis of vehicle traffic, the general demand is to compute how many times vehicles pass through a certain traffic checkpoint, on a certain road, and in an entire region. For vehicle tracking, the general demand is to extract location information about all checkpoints that vehicles have passed within a certain time period from the massive

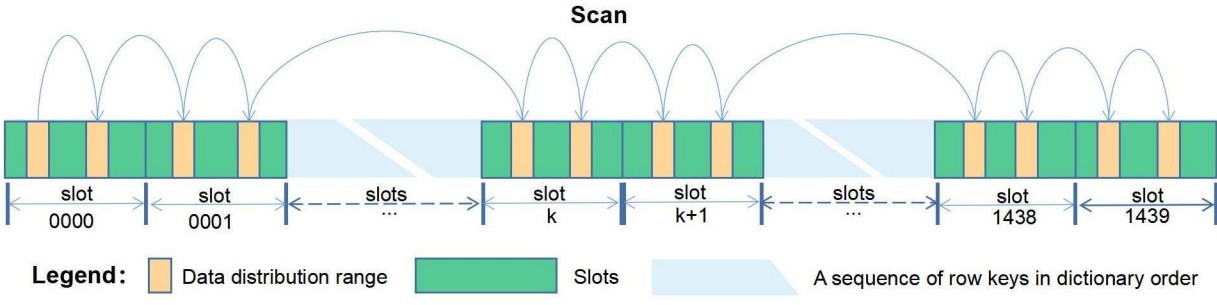

**Fig 4. The location of the slice data within the time range.**

VLPD. Both the statistical analyses for multiple conditions and the path tracking for vehicles need to be meaningful within a specific time frame. In other words, for all kinds of VLPD applications, the time factor is a necessary condition.

Based on this categorization, the time period information in the query request for all VLDP application problems can be used to determine the distribution range of the query data in the line key sequence. After receiving a user query request, the client determines the data distribution range as follows:

- Extract the time span information;

- Slice the time period and calculate the slot where the slice is located;

- Combine the slot number and time slice to form a line key prefix;

- Build a scan-task stack in which each element (scan) contains a row key prefix.

Fig 4 shows a schematic representation of the position of the slice data in the row key sequence for a certain time range.

As can be seen in Fig 4, the data is discrete and evenly distributed over the entire sequence of row keys. Using the row key prefixes defined by the combinations of slot numbers and time slices, the distribution range of the data can be precisely defined. The data scan range generation algorithm can be described by the following Java code.

```
Algorithm 3: Data Distribution Range Generation Algorithm
1. private Stack<Scan> getScans(Long fromStamp,Long toStamp){
2.      Stack<Scan> scans = new Stack<>();
3.      for(Long l = fromStamp;l < = toStamp;l + = 60000){
4.        Scan scan = new Scan();
5.        Timestamp timestamp = new Timestamp(l);
6.        DateFormat sdf = new SimpleDateFormat("yyyy-MM-dd HH:mm:
ss");
7.        String dateStr = sdf.format(timestamp);
8.        String prefix = String.valueOf(ch(l,numServers)+10000).sub-
string(1) +"|" +
                dateStr.substring(0,16);
9.        scan.setRowPrefixFilter(prefix.getBytes());
10.        scans.push(scan);
11.      }
12.    return scans;
13.  }
```

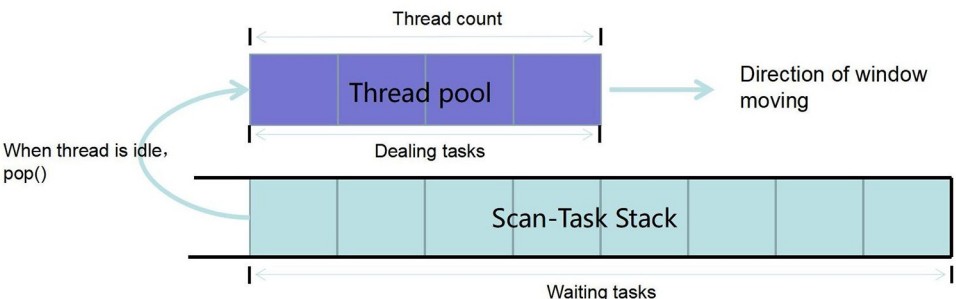

**Fig 5. Multi-threaded sliding window for parallel requests.**

However, since the data is discretely distributed over the row key sequence, this creates a problem for data scanning queries. Obviously, to improve the query efficiency, the entire sequence of row keys should not be scanned from beginning to end, but rather, the scan should jump forward (as shown in Fig 4). For this reason, there is a need to design a parallel algorithm that can support jump scanning in order to improve the scanning efficiency.

## 4.2 Multi-threaded sliding window algorithm for a parallel request

Big data platforms are equipped with extreme parallel data processing capabilities. Using parallel algorithms to process multiple scan tasks can significantly improve the query efficiency. Suppose that $T$ scan tasks have been constructed based on the time period in the query request and that these tasks are stored in the scan-task stack. The algorithm for the parallel processing of the scan tasks in the stack can be described as follows:

- Initialize a process pool with a window width of *n*;

- For each process in the window, repeatedly perform a stack-out operation on the scan-task stack and the corresponding scan operation;

- End the thread when the pop() return value is empty;

- Print the scan result.

The algorithm execution process is illustrated in Fig 5.

In this algorithm, the thread count (the window width) needs to be determined jointly with the client-side performance and the number of data node servers on the server side (empirical parameters for determining the window width value are given in the experimental section 5). As the number of scan tasks in the stack keeps decreasing, the process pool window keeps sliding to the right until the stack is empty, ending all processes and completing the query. The algorithm can be further described by the following Java code.

```
Algorithm 4: Multi-threaded Sliding Window for Parallel Request
Algorithm
1. private Result scanConcurrentlySlidWindow(Table table,Stack<Scan>
scans) {
2.      Result result = new Result();
3.      ExecutorService executor = Executors.newFixedThreadPool
(numThreads);
4.      CountDownLatch latch = new CountDownLatch(numThreads);
5.      for(int I = 0;I < numThreads;i++){
6.        executor.submit(()->{
```

```
7.        ResultScanner scanner;
8.        while (!scans.empty()){
9.          Scan scan = scans.pop();
10.           scanner = table.getScanner(scan);
11.           for (Result rs:scanner){
12.             //get the result
13.             result.merge(rs);
14.           }
15.           scanner.close();
16.         }
17.         latch.countDown();
18.       });
19.     }
20.     latch.await();
21.     executor.shutdown();
22.     return result;
23. }
```

In this code, the method *scanConcurrentlySlidWindow* receives two parameters, an HBase large table object and a scan-task stack, and it returns the query result after the method execution is completed. In the method, the for-loop initializes *numThreads* threads for the parallel processing of the scan tasks. When there are no new tasks to be processed, the threads are terminated.

## 5. Performance analysis

In order to verify the performance of the VLPD access algorithm described in this paper in practice, we built a highly available Hadoop environment on a physical cluster with five physical nodes and installed and ran an HBase database on top of it to conduct a comprehensive test analysis of the uniformity of the data distribution and the data access efficiency on the server.

### 5.1 Physical hardware and application processes

The physical servers in the cluster are divided into two categories, NameNode servers and DataNode servers. Servers running NameNode are allocated more memory and fewer storage resources, while servers running DataNode are allocated more storage and fewer memory resources. The specific hardware resource allocations are shown in Table 1.

### 5.2 Analysis of even data distribution

The uniformity of data distribution can be measured by the variance of the number of rows distributed on the region server. About 29 million rows VLPD for the entire month of April 2021 in the Tianjin Sino-Singapore Eco-city were selected as the experiment sample, and the total amount of data in the slots was obtained by counting the number of data rows with the same slot number in the row key header. The distribution of the VLPD clustered on the region servers is shown in Table 2.

Using the assigned region number on the region server as the horizontal axis and the number of rows of VLPD saved on the different region server as the vertical axis, a histogram of the data on the region server was drawn (as shown in Fig 6).

We used the variance equation

$$s^2 = \frac{\sum_{i=1}^{n} (x_i - x)^2}{n} \tag{3}$$

**Table 1. Experimental hardware resource allocation.**

| Server | Memory | Hard disk | Hadoop (high-availability) process | Hbaseprocess |
|---|---|---|---|---|
| n1 | 8G | 200G | Namenode: DFSZKFailoverController | HMaster |
| n2 | 8G | 200G | Namenode: DFSZKFailoverController | HMaster |
| d1 | 4G | 500G | JournalNode: QuorumPeerMain | Datanode、HRegionServer |
| d2 | 4G | 500G | JournalNode: QuorumPeerMain | Datanode、HRegionServer |
| d3 | 4G | 500G | JournalNode: QuorumPeerMain | Datanode、HRegionServer |

to compute the variance value, and the variance value(40.3) is very low compared to the average value(1216.1).

It can be seen that the data pieces of VLPD, which has been divided into slices by "minutes", have been evenly distributed over the region server. The read/write hotspot problem has thus been effectively avoided by the storage model based on JHCA.

## 5.3 Analysis of data retrieval efficiency

The analysis of the comprehensive query conditions of VLPD can all be achieved by scanning the data within a certain time period. Therefore, the system's efficiency in processing data retrieval query requests can be measured by the length of time needed for the data scanning operations made by the cluster. Based on the data access algorithm model described in this

**Table 2. Distribution of VLPD on region servers.**

| Region Server | | Row Key Starter | Row Key Terminator | Data Volumne (rows×1000) | Average | Variance |
|---|---|---|---|---|---|---|
| No. | Partiton | | | | | |
| 1 | d1:1 | | 60 | 1253.956 | 1216.1 | 40.3 |
| 2 | d2:1 | 60 | 120 | 1213.48 | | |
| 3 | d3:1 | 120 | 180 | 1157.154 | | |
| 4 | d1:2 | 180 | 240 | 1269.611 | | |
| 5 | d2:2 | 240 | 300 | 1265.614 | | |
| 6 | d3:2 | 300 | 360 | 1214.047 | | |
| 7 | d1:3 | 360 | 420 | 1276.413 | | |
| 8 | d2:3 | 420 | 480 | 1238.357 | | |
| 9 | d3:3 | 480 | 540 | 1224.060 | | |
| 10 | d1:4 | 540 | 600 | 1200.573 | | |
| 11 | d2:4 | 600 | 660 | 1245.171 | | |
| 12 | d3:4 | 660 | 720 | 1227.063 | | |
| 13 | d1:5 | 720 | 780 | 1247.234 | | |
| 14 | d2:5 | 780 | 840 | 1187.653 | | |
| 15 | d3:5 | 840 | 900 | 1220.808 | | |
| 16 | d1:6 | 900 | 960 | 1202.009 | | |
| 17 | d2:6 | 960 | 1020 | 1192.463 | | |
| 18 | d3:6 | 1020 | 1080 | 1234.996 | | |
| 19 | d1:7 | 1080 | 1140 | 1123.572 | | |
| 20 | d2:7 | 1140 | 1200 | 1174.042 | | |
| 21 | d3:7 | 1200 | 1260 | 1231.612 | | |
| 22 | d1:8 | 1260 | 1320 | 1138.808 | | |
| 23 | d2:8 | 1320 | 1380 | 1267.322 | | |
| 24 | d3:8 | 1380 | | 1179.604 | | |

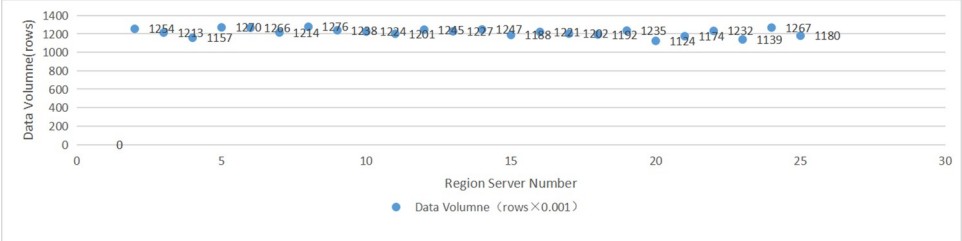

**Fig 6. Data distribution on region server.**

paper, the factors affecting the data scanning efficiency include both the parallel scanning window width (the number of parallel request threads) and the query data time range.

**(1)Analysis of scan time with different window widths.** Using sliding window widths ranging from 1 to 12, VLPDs with time spans of 1, 2, and 3 days were scanned separately, and the time taken to complete each scan was recorded. The experimental results are shown in Fig 7.

The experiments show that when the sliding window width is set to 1–3, the time used for scanning varies the most; and when the sliding window width is greater than 3, the effect on the scanning time is not very obvious.

The window width of 3 coincides with the number of data nodes in the experimental cluster. Optimal performance results can be obtained by setting the width of the moving window to the number of physical data node servers in the cluster when implementing access to large data from traffic car crossings using the traffic car data algorithm model described in this paper.

**(1) Comparative analysis of query efficiency for different data scales.** A comparative analysis of the time taken for the same query condition with different data sizes can visually reflect the algorithm's execution efficiency. This paper compares the differences in query efficiency between the two types of storage solutions by storing the day-by-day cumulative sample VLPD in the HBase database and in the popular PostgreSQL database under the condition of performing the same query function operation. In the experiments, we use the data for the April 2021 VLPD in the China-Singapore Tianjin Eco-city as the experimental sample data, and we determine the time spent on querying the vehicle with license plate number "津 AYH6**" on April 1 by accumulating the data storage size of the PostgreSQL database and the HBase database day by day. For the PostgreSQL database, we used the following script:

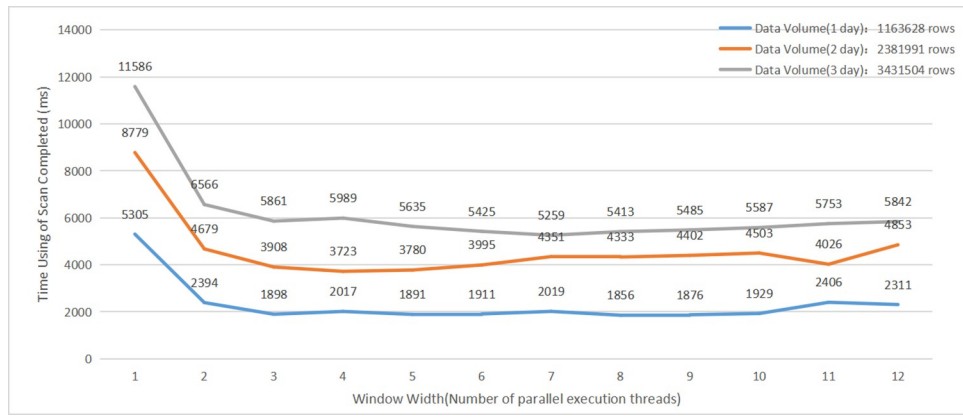

**Fig 7. Impact of concurrent scan operations on query efficiency.**

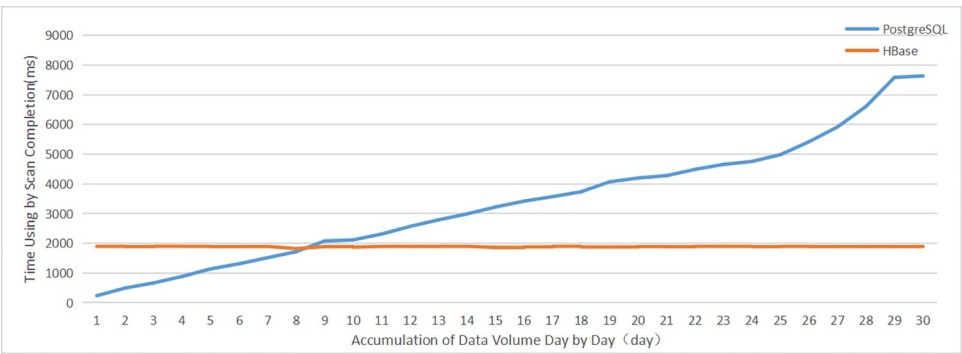

**Fig 8. Comparative analysis of server performance using different solutions.**

```
select * from vehicle t where t. "HPHM" = '津AYH6**' and t. "JGSK"
BETWEEN to_timestamp('2021-04-01 00:00:00',
' YYYY-MM-DD HH24:MI:SS') and to_timestamp('2021-04-01
23:59:59','YYYY-MM-DD HH24:MI:SS');
```

to obtain the query result(Considering the privacy issue, we hide the last two digits of the license plate number in the sql script). For the HBase database, we used the algorithm described in this paper to perform a scan operation on the same volume data size. The query result was obtained via a filter. The experimental results are shown in Fig 8.

The experiments show that both PostgreSQL and HBase return the same result with a total row count of 28. For the PostgreSQL database, the time used for querying increases linearly with the accumulation of the data volume day by day, so that the time complexity is O($n$), while for the HBase database, which accesses data using the algorithm described in this paper, the data size has no significant effect on the query efficiency, giving a time complexity of O(1). Although under the condition of a small data size (cumulative data for 8 days, fewer than $1\times10^{7}$ rows), use of the PostgreSQL database to store the VLPD provides better performance. However, as the accumulated data in the database increases, the query time used by the PostgreSQL database is significantly higher than that used by the HBase database, and eventually it cannot complete the demand for real-time data retrieval.

## 5.4 Characteristics comparison

By comparing and analyzing the research results of current VLPD access models, this paper summarizes the characteristics of different models from the perspectives of whether they support multi-conditioned queries, space usage, memory consumption, horizontal scalability, and real-time access. Compared with models based on auxiliary indexing technology, MapReduce, and Spark, JHAC based VLPD access model has the characteristics of fully supporting multi condition comprehensive queries, saving storage space, reducing memory consumption, supporting horizontal scalability, and meeting real-time data access. Table 3 presents the comparative analysis results of the characteristics of different models.

## 6. Conclusions

To address the problem of massive VLPD storage and applications in the field of intelligent transportation, this paper proposes a multi-threaded sliding window parallel access model based on the JHCA. The model guarantees a load balance for data access by slicing the VLPD by "minutes" (of the day) and using JHCA to evenly distribute the sliced data in the preset slots

**Table 3. Characteristics comparison among different models.**

| Models | Characteristics | | | | |
|---|---|---|---|---|---|
| | Multi-conditioned Query | Space usage | Memory consumption | Scalability | Real-time Access |
| auxiliary indexing | partly | wasteful | Saving | supported | supported |
| MapReduce based | supported | Saving | Saving | supported | No |
| Spark based | partly | Saving | wasteful | No | supported |
| JHAC based | supported | saving | Saving | supported | supported |

of a big data platform. The data reading model based on a multi-threaded sliding window is designed to realize high-speed scanning of data within a specified time range by taking advantage of the high efficiency of the big data platform in handling concurrent requests, thus realizing relatively efficient multi-conditional comprehensive query and data analysis on the key–value database environment of a big data platform.

The VLPD access model designed in this paper is based on a native key–value database, without using any additional third-party engines. The algorithm model has outstanding characteristics such as supporting multi-condition query fully, saving storage space, low memory usage, efficiency, scalability, and can meet the needs of real-time data access. Just as every coin has tow sides, this model also has limitations. Through the scan operation, the VLPD located in a slot must be extracted to the client for the further analysis. Compared to the model based on auxiliary indexing, the network load is heaver. In future research, we can explore other types of traffic data processing algorithms and further optimize the algorithm model proposed in this paper to solve the problem of high network bandwidth consumption and meet the needs of different application scenarios.

The algorithm model described in this paper has been successfully applied to the "Tianjin Sino-Singapore Eco-city High Performance Big Data Platform" project. Application scenarios, such as vehicle trajectory query and in-time traffic flow analysis, shows that the proposed data access model is flexible and efficient, simple to implement, easy to deploy, and easy to maintain, and thus worth promoting in the development and utilization of massive VLPD in the fields of smart cities and smart transportation.

## Supporting information

**S1 Data.**
(ZIP)

## Author Contributions

**Conceptualization:** Wei Wang.

**Data curation:** Wenfang Cheng, Yuran Zhang, Yingfang Yu.

**Methodology:** Wei Wang.

**Resources:** Jing Chen.

**Software:** Wei Wang.

**Validation:** Wenfang Cheng, Zhen Wang.

**Writing – original draft:** Wei Wang.

**Writing – review & editing:** Jing Chen, Zhen Wang.

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
