## [Decision Letter · Decision Letter 0]

16 May 2023

PONE-D-23-10217A Vehicle License Plate Data Access Model Based on the Jump Hash Consistency AlgorithmPLOS ONE

Dear Dr. wang,

Thank you for submitting your manuscript to PLOS ONE. After careful consideration, we feel that it has merit but does not fully meet PLOS ONE’s publication criteria as it currently stands. Therefore, we invite you to submit a revised version of the manuscript that addresses the points raised during the review process.

We look forward to receiving your revised manuscript.

Kind regards,

Ayesha Maqbool, PhD

Academic Editor

PLOS ONE

Journal Requirements:

Reviewers' comments:

Reviewer's Responses to Questions

**Comments to the Author**

1. Is the manuscript technically sound, and do the data support the conclusions?

Reviewer #1: Partly

Reviewer #2: Yes

Reviewer #3: Yes

2. Has the statistical analysis been performed appropriately and rigorously? 

Reviewer #1: Yes

Reviewer #2: No

Reviewer #3: Yes

3. Have the authors made all data underlying the findings in their manuscript fully available?

Reviewer #1: Yes

Reviewer #2: No

Reviewer #3: Yes

4. Is the manuscript presented in an intelligible fashion and written in standard English?

Reviewer #1: Yes

Reviewer #2: Yes

Reviewer #3: Yes

5. Review Comments to the Author

Reviewer #1: The main purpose of this paper is to address the problems of implementing complex multi-conditional comprehensive query functions and flexible data applications in the key–value database storage environment of a big data platform, this paper proposes a data access model based on the jump hash consistency algorithm. Algorithms such as data slice storage and multi-threaded sliding window parallel reading are used to realize evenly distributed storage and fast reading of massive time-series data on clustered data nodes. However, there are some issues needing to be solved.

1. In “Algorithm 1”, is the “Algorithm 1” correct? Operator symbol “>>>” ?

2. In Figure 8, the data label “Hbase” should be “HBase”.

3. The abbreviation format of nouns in the text is not uniform, for example: linear congruence method (LCG), jump consistent hash algorithm (JCHA).

4. You’d better figure out the exact uniform variance of the data distribution in your experiment, and list the corresponding variance criteria.

5. The experimental part should increase the comparison with traditional methods in performance and characteristics, showing advantages and breakthroughs.

6. It is suggested to increase your comparison model, especially the model in recent years, which can increase persuasion of the paper’s method.

7. Some important references are missed. The authors should analyze more new related works. E.g. A Federated Learning-based License Plate Recognition Scheme for 5G-enabled Internet of Vehicles, IEEE Transactions on Industrial Informatics 17, no.(12) , pp.8523–8530, 2021

Reviewer #2: Paper has the potential to be published after addressing following problems.

1. Paper is well written but overall proofreading is required to avoid minor spelling/typing mistakes.

2. Authors should justify the novelty of their work.

3. Results should be compared with some state of the art technique.

Reviewer #3: This article proposes a multi-threaded sliding window parallel access model based on Jump Consistent Hash Algorithm to address the storage and application issues of massive vehicle trajectory data (VLPD) in the intelligent transportation field. The model effectively avoids cluster hotspots, supports comprehensive queries with complex conditions, and data analysis. In addition, the model has been successfully applied in practical projects for vehicle trajectory queries and traffic flow analysis, proving its high application value in the development and utilization of massive vehicle trajectory data in smart cities and intelligent transportation fields. Overall, the article has a reasonable structure, clear logic, and high completeness. I recommend this paper for publication.

My review comments are as follows:

1.In the introduction section, the current status and development trends of intelligent transportation systems can be further introduced, and the application and value of license plate data generated by intelligent transportation systems can be described in more detail.

2.In the experimental section, more detailed experimental details and parameter settings can be provided so that readers can replicate the experiment and verify the results. Verification of the correctness of query results can be provided, for example, by comparing query results with manually counted results or query results from other databases.

3.In the conclusion, the limitations of the algorithm can be added, and future research directions can be discussed. For example, the algorithm model has efficiency and scalability, and can meet the needs of real-time data access. In future research, we can explore other types of traffic data processing algorithms and further optimize the algorithm model proposed in this paper to meet the needs of different application scenarios.

6. PLOS authors have the option to publish the peer review history of their article (what does this mean?). If published, this will include your full peer review and any attached files.

Reviewer #1: No

Reviewer #2: No

Reviewer #3: No

---

## [Author Response · Author response to Decision Letter 0]

9 Jun 2023

Response to reviewer1:

Question 1:  In “Algorithm 1”, is the “Algorithm 1” correct? Operator symbol “>>>” ?

Answer: Yes, it is right. “>>>” means an unsigned right shift. The experimental results indicate that there is no problem here.

Question 2: In Figure 8, the data label “Hbase” should be “HBase”.

Answer: Thank you. I have revised it.

Question 3: The abbreviation format of nouns in the text is not uniform, for example: linear congruence method (LCG), jump consistent hash algorithm (JCHA).

Answer: Thank you. I have carefully checked to ensure consistency in naming conventions.

Question 4: You’d better figure out the exact uniform variance of the data distribution in your experiment, and list the corresponding variance criteria.

Answer: Thanks for your suggestion. I give the variance of the data distribution in my experiment by table 2.

Question 5: The experimental part should increase the comparison with traditional methods in performance and characteristics, showing advantages and breakthroughs.

Answer: I Added section 5.4 in the manuscript. In this section, a qualitative comparison analysis was conducted with traditional data access models from the perspectives of whether it supports multi conditional queries, disk usage, memory consumption, horizontal scalability, and whether it supports real-time access.

Question 6: It is suggested to increase your comparison model, especially the model in recent years, which can increase persuasion of the paper’s method.

Answer: Added qualitative comparison analysis to the models in recent years, such as based on auxiliary index technology models, based on MapReduce technology models, and based on Spark technology models in the article.

Question 7: Some important references are missed. The authors should analyze more new related works. E.g. A Federated Learning-based License Plate Recognition Scheme for 5G-enabled Internet of Vehicles, IEEE Transactions on Industrial Informatics 17, no.(12) , pp.8523–8530, 2021

Answer: Thanks for you suggestion. I added references to the latest research findings.

Response to reviewer2:

Question 1: Paper is well written but overall proofreading is required to avoid minor spelling/typing mistakes.

Answer: Thanks for your careful review. I have carefully checked and corrected the spelling/typing mistakes.

Question 2: Authors should justify the novelty of their work.

Answer: In the section of result, I emphasized the novelty of this work. It includes: supporting multi-condition query fully, saving storage space, low memory usage, efficiency, scalability, and can meet the needs of real-time data access.

Question 3: Results should be compared with some state of the art technique.

Answer: I Added section 5.4 in the manuscript. In this section, a qualitative comparison analysis was conducted with traditional data access models from the perspectives of whether it supports multi conditional queries, disk usage, memory consumption, horizontal scalability, and whether it supports real-time access.

Response to reviewer3:

Question 1. In the introduction section, the current status and development trends of intelligent transportation systems can be further introduced, and the application and value of license plate data generated by intelligent transportation systems can be described in more detail.

Answer: Thanks for your suggestions. I have added content about trends in ITS and some detailed information about the value of license plate data.

Question 2: In the experimental section, more detailed experimental details and parameter settings can be provided so that readers can replicate the experiment and verify the results. Verification of the correctness of query results can be provided, for example, by comparing query results with manually counted results or query results from other databases.

Answer: I listed detail result data in table 2 and added a note in this paper about the consistency of the experimental results. Thank you.

Question 3: In the conclusion, the limitations of the algorithm can be added, and future research directions can be discussed. For example, the algorithm model has efficiency and scalability, and can meet the needs of real-time data access. In future research, we can explore other types of traffic data processing algorithms and further optimize the algorithm model proposed in this paper to meet the needs of different application scenarios.

Answer: Thanks for your great suggestion. The conclusion section is further refined by adding the strengths and limitations of the algorithms described in this paper and directions for future research.

---

## [Decision Letter · Decision Letter 1]

29 Jun 2023

A Vehicle License Plate Data Access Model Based on the Jump Hash Consistency Algorithm

PONE-D-23-10217R1

Dear Dr. Wang,

We’re pleased to inform you that your manuscript has been judged scientifically suitable for publication and will be formally accepted for publication once it meets all outstanding technical requirements.

Kind regards,

Ayesha Maqbool, PhD

Academic Editor

PLOS ONE

Additional Editor Comments (optional):

Reviewers' comments:

Reviewer's Responses to Questions

**Comments to the Author**

1. If the authors have adequately addressed your comments raised in a previous round of review and you feel that this manuscript is now acceptable for publication, you may indicate that here to bypass the “Comments to the Author” section, enter your conflict of interest statement in the “Confidential to Editor” section, and submit your "Accept" recommendation.

Reviewer #1: (No Response)

Reviewer #3: All comments have been addressed

2. Is the manuscript technically sound, and do the data support the conclusions?

Reviewer #1: (No Response)

Reviewer #3: Yes

3. Has the statistical analysis been performed appropriately and rigorously? 

Reviewer #1: (No Response)

Reviewer #3: Yes

4. Have the authors made all data underlying the findings in their manuscript fully available?

Reviewer #1: (No Response)

Reviewer #3: Yes

5. Is the manuscript presented in an intelligible fashion and written in standard English?

Reviewer #1: (No Response)

Reviewer #3: Yes

6. Review Comments to the Author

Reviewer #1: (No Response)

Reviewer #3: The author has made the revisions according to previous comments, and I suggest accepting this paper.

7. PLOS authors have the option to publish the peer review history of their article (what does this mean?). If published, this will include your full peer review and any attached files.

Reviewer #1: No

Reviewer #3: No

---

## [Editor Report · Acceptance letter]

16 Aug 2023

PONE-D-23-10217R1 

A Vehicle License Plate Data Access Model Based on the Jump Hash Consistency Algorithm 

Dear Dr. Cheng:

I'm pleased to inform you that your manuscript has been deemed suitable for publication in PLOS ONE. Congratulations! Your manuscript is now with our production department. 

Kind regards, 

on behalf of

Dr. Ayesha Maqbool 

Academic Editor

PLOS ONE